# COUNTERING LANGUAGE DRIFT VIA GROUNDING

## ABSTRACT

While reinforcement learning (RL) shows a lot of promise for natural language processing—e.g. when fine-tuning natural language systems for optimizing a certain objective—there has been little investigation into potential *language drift*: when an external reward is used to train a system, the agents' communication protocol may easily and radically diverge from natural language. By re-casting translation as a communication game, we show that language drift indeed happens when pre-trained agents are fine-tuned with policy gradient methods. We contend that simply adding a "naturalness" constraint to the reward, e.g. by using language model log likelihood, does not fully address the issue, and argue that (perceptual) *grounding* is required. That is, while language model constraints impose syntactic conformity, they do not lead to semantic correspondence. Our experiments show that grounded models give the best communication performance, while retaining English syntax along with the ability to convey the intended semantics.

## 1 INTRODUCTION

In the summer of 2017, the internet was briefly abuzz with the mistaken viral message that a leading AI research lab had to "unplug its AI" because it "had gone rogue". What had in fact happened was that two chatbots, under certain conditions, had, rather unsurprisingly, started diverging from their English training data and had instead reverted to their own ungrammatical communication protocol for solving a negotiation task (Lewis et al., 2017). Instead of saying something like "hats have no value for me" the system would starting saying things like "hat have zero to me to me to me to me". As was soon made clear by the parties involved, this sort of language drift is to be expected if we are optimizing for an external reward, for example one based on whether or not two agents successfully accomplish a negotiation.

While language drift is to be expected under external reward, it is natural to ask what we can do to avoid it. Consider policy gradient methods, for example, and suppose we sample an output sequence from an English language decoder for a given task: sampling a non-grammatical sequence might still be rewarded if we manage to solve the task (e.g., due to some correct words; or because an interlocutor understood us anyway, or guessed correctly), which would quickly move the decoder away (drifting) from English. If we were able to keep drift in check, we could maximize reward while *retaining* the "Englishness" of the decoder, with obvious benefits for interpretability and interaction with humans. That is, while search space size prohibits the direct usage of policy gradient methods for training natural language decoders from scratch, we could prevent pre-trained models from drifting while we optimize for the desired reward using policy gradients.

Thus, the ability to stop policy gradient methods from diverging from natural language enables interesting long-term possibilities for exploration: imagine e.g. fine-tuning a pre-trained language model trained on large amounts of data, call it a "language module", for a given generation task with limited data. When training chit-chat dialogue agents, for example, we often want to optimize for some very high-level reward, such as engagingness or consistency, with hardly enough data to learn simple English grammar. Or consider what might happen when we train agents using self-play to *actively* use natural language to change the other agent's (mental) state, rather than having a model passively observe language usage in some corpus or dataset, as usually happens.

In this work, we study the question of language drift. Drawing inspiration from Lee et al. (2018), we re-cast translation as a communication game. Two machine translation (MT) agents—i.e., encoder-decoder models with attention—are tasked with successfully translating source language sequences

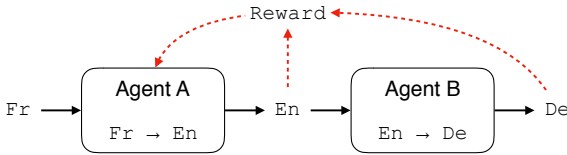

Figure 1: Diagram of our communication game.

to the target language using a third pivot language as an intermediary. The communication channel (the output of the first agent's decoder, which is fed to the second agent's encoder as input) is updated via policy gradient methods to optimize for translating into the target language, effectively fine-tuning two separate pre-trained MT models via a pivot language.

While the subject of communication need not be language (e.g., for Lee et al. (2018), agents learn to translate by communicating about images), three-way translation via pivot is an excellent way for studying the current problem: we can check exactly to what extent the communicated sequence corresponds to both the intended meaning, as well as to the gold standard sequence. We can think of this setup as an agent aiming to communicate its state to another agent via some protocol—yet in this case, the mental states and intermediary communication protocol are completely interpretable.

In what follows, we show that language drift happens, and quite dramatically so, when fine-tuning using policy gradients. We then show that the most intuitive way of solving this problem—adding an "Englishness" constraint, such as the log-probability assigned by a language model, to the reward function—does not in fact lead to the desired consequences. Indeed, there is nothing preventing such models from learning to translate "Two giraffes standing next to a white truck in the savanna" from French to German via "Democracy is a political system" as the English intermediary.

Hence, we contend that what is missing is *grounding*: while language model constraints impose syntactic conformity, they do not lead to semantic correspondence. Humans don't invent unique idiolects for every individual interlocutor, exactly because they are grounded: we share strong priors, social and behavioral norms, and a common sensorimotor experience of our physical environment. Thus, "not going rogue" means not only sticking to the prescribed language, but more importantly preserving meaning, which means staying grounded.

Our experiments show that fine-tuning the communication channel with visual grounding leads to the highest communication performance (Fr→En→De) as well as the best retention of original syntax and intended semantics. Our token frequency analysis corroborates our hypothesis, and shows that grounding is key for preserving the token frequency distribution of the pivot language (English).

## 2 PRIOR WORK

Our work is inspired by recent work in protocols or languages that emerge from multi-agent interaction (Lazaridou et al., 2017; Lee et al., 2018; Andreas et al., 2017; Evtimova et al., 2018; Kottur et al., 2017; Havrylov & Titov, 2017; Mordatch & Abbeel, 2017). Work on the emergence of language in multi-agent settings goes back a long way (Steels, 1997; Nowak & Krakauer, 1999; Kirby, 2001; Briscoe, 2002; Skyrms, 2010). In our case, we are specifically interested in *tabula inscripta* agents that are already pre-trained to generate natural language, and we are primarily concerned with keeping their language natural during further training.

Reinforcement Learning (RL) has been applied to fine-tuning models for various natural language generation tasks, including summarization (Ranzato et al., 2015; Paulus et al., 2017), information retrieval (Nogueira & Cho, 2017), MT (Gu et al., 2017; Bahdanau et al., 2016) and dialogue (Li et al., 2017). Our work can be viewed as fine-tuning MT systems using an intermediary pivot language. In MT, there is a long line of work of pivot-based approaches, most notably Muraki (1986) and more recently with neural approaches (Wang et al., 2017; Cheng et al., 2017; Chen et al., 2018). There has also been work on using visual pivots directly (Hitschler et al., 2016; Nakayama & Nishida, 2017; Lee et al., 2018). Grounded language learning in general has been shown to give significant practical improvements in various natural language understanding tasks (Gella et al., 2017; Elliott & Kádár, 2017; Chrupała et al., 2015; Kiela et al., 2017; Kádár et al., 2018). Meanwhile, Bowman et al. (2016) found a powerful decoder to ignore the latent representation in VAEs for language.

## 3 TASK AND MODELS

We recast translation as a communication game involving two MT agents: Fr→En and En→De (see Figure 1). Our dataset consists of $N$ triples of aligned sentences $\{\text{Fr}_i, \text{En}_i, \text{De}_i\}_{i=1}^N$, where $\text{En}_i$ is only used for evaluation. We first feed the French sentence $\text{Fr}_i$ to Agent A, which generates an English message $\overline{\text{En}_i}$ as output. Agent B is then trained to maximize the log likelihood of the ground truth German sentence given the English message, i.e. $\log p(\text{De}_i | \overline{\text{En}_i})$. Agent A is trained using REINFORCE (Williams, 1992) with reward $R = \log p_B(\text{De}_i | \overline{\text{En}_i})$.[1] This encourages Agent A to develop helpful communication policies for Agent B, and allows Agent B to adapt to Agent A's new policies. In other words: communication via the pivot language (English) is a success if we are able to translate the intended source sequence (French) into the desired target sequence (German).

Both agents are pre-trained individually before communication, meaning that we start off with English as an intermediate language in the early stages of the game. This work examines what happens to the intermediate language as we fine-tune the system jointly: will the agents keep communicating in English, or diverge? And if so, what can we do to prevent that from happening?

### 3.1 AUXILIARY TASKS

To help reduce the search space of intermediate languages, we use two auxiliary tasks: language modelling (LM) and image-caption retrieval (henceforth called the grounding model).

**Language Model** Given a language model pre-trained on a standard English corpus, the log likelihood of the English message informs its general "Englishness". We incorporate this into the reward for Agent A, so that it learns to send messages that are plausible English.[2] Reward for Agent A is:

$$R_{\text{LM}} = \log p_B(\text{De}_i | \overline{\text{En}_i}) + \beta_{LM} \log p_{LM}(\overline{\text{En}_i}).$$

**Grounding Model** Let us assume we have access to a set of images $\{\text{Img}_i\}$ associated with each triple $\{\text{Fr}_i, \text{En}_i, \text{De}_i\}$. Given a pre-trained image-caption retrieval model, such as VSE++ (Faghri et al., 2018), the log likelihood of the image given the English message (and vice versa) informs how much the English message is grounded in the original semantic content (Kiela et al., 2017). We incorporate the ranking loss into Agent A's reward.

$$R_{\text{G}} = \log p_B(\text{De}_i | \overline{\text{En}_i}) + \beta_G \log p_G(\text{Img}_i | \overline{\text{En}_i}).$$

Note that $\beta_{LM}, \beta_G$ are hyperparameters.

### 3.2 TRAINING OBJECTIVE

For brevity the $t$-th token in the $i$-th English sentence $\text{En}_{i;t}$ is abbreviated to $\text{En}_t$, and $\text{En}_i$ to En.

**Policy Gradient Training** At decoding timestep $t$, Agent A takes an action (outputs token $\overline{\text{En}_t}$) given an environment (previous hidden states and previous token $\overline{\text{En}_{t-1}}$). It receives reward $R$ at the end of the sequence, from which we subtract a state-dependent baseline $\overline{R_t}$ to reduce variance. Therefore, we maximize $(R - \overline{R_t}) \log p(\overline{\text{En}_t} | \overline{\text{En}_{<t}}, \text{Fr})$. In addition, we employ entropy regularization on Agent A's decoder to encourage exploration. Hence, Agent A's overall objective function is given as:

$$\mathbb{L}_A = \alpha_{\text{pg}}(R - \overline{R_t}) \log p(\overline{\text{En}_t} | \overline{\text{En}_{<t}}, \text{Fr}) + \alpha_{\text{entr}} H(p(\overline{\text{En}_t} | \overline{\text{En}_{<t}}, \text{Fr})) - \alpha_{\text{b}} \text{MSE}(R, \overline{R_t}),$$

where $H$ and MSE denote entropy and mean squared error losses.

**Cross Entropy Training** Agent B is trained using standard cross entropy loss, i.e.

$$\mathbb{L}_B = \log p(\text{De}_t | \text{De}_{<t}, \overline{\text{En}}).$$

We jointly train both agents by maximizing $\mathbb{L} = \mathbb{L}_A + \mathbb{L}_B$.

---

[1] We use subscript $B$ to denote that the probability is computed with Agent B.

[2] We also experimented with a dense LM reward on the word-level, but found this to lead to worse performance. We hypothesize that the model might be focusing too much on the dense LM reward, ignoring the sparse reward for the communication task and leading to poor performance.

## 4 EXPERIMENTAL SETTINGS

In this section we provide the details of our experimental setup: a Fr→X→De translation task where the intermediate language X is initialized as English, and subsequently fine-tuned with policy gradient. On a trilingual corpus consisting of three languages (Fr, En and De), we can measure communication success with Fr→De BLEU (Papineni et al., 2002), while Fr→En BLEU informs how closely the intermediate language resembles English, at any given point during fine-tuning.

**Datasets** Agents are initially pre-trained on IWSLT Fr→En and En→De. Fine-tuning is performed on Multi30k Task 1(Elliott et al., 2016). That is, importantly, there is no overlap in the pre-training data and the fine-tuning data. Multi30k Task 1 consists of 30k images and one caption per image in English, French, German and Czech (of which we only use the first three). For the English language model, we compare four different datasets: WikiText103, MS COCO and Flickr30k. The image-caption retrieval model is trained on Flickr30k: the same set of 30k images as Multi30k but containing 5 English captions per image. Following Faghri et al. (2018), we randomly crop training images at every epoch. We use 2048-dimensional final-layer features from a pretrained and fixed ResNet-152 (He et al., 2016).

**Preprocessing** The same tokenization and vocabulary are used across different tasks and datasets. We lowercase and tokenize our corpora with Moses (Koehn et al., 2007) and use subword tokenization with Byte Pair Encoding (BPE) (Sennrich et al., 2016) with 10k merge operations. This allows us to use the same vocabulary across different models seamlessly (translation, language model, image-caption ranker model).

**Controlling the English message length** When fine-tuning the agents, we observe that the length of English messages becomes excessively long. As Agent A has no explicit incentive to output the ⟨EOS⟩ symbol, it tends to keep transmitting the same token repeatedly. Excessively long messages obscure evaluation of the communication protocol. For instance, BLEU score quickly deteriorates as the message length becomes longer, as it is a precision metric. When the message length is fixed, a drop in BLEU score will by necessity mean that the intermediate language has drifted away more. For this reason, we constrain the length of English messages to be no longer than the length of their French source sentence, or shorter if the model outputs the ⟨EOS⟩ symbol early. Recall that Agent B is supervised to predict the ⟨EOS⟩ symbol, so does not suffer from this issue.

**Model Architecture and Pretraining** Our MT agents are standard sequence-to-sequence models with attention (Bahdanau et al., 2015) with unidirectional, 1-layer GRU with 256 hidden units and 256-dimensional embeddings. During initial pre-training on IWSLT, we early-stop based on BLEU score on the development set (tst2013). The best checkpoints give 34.05 BLEU and 21.94 BLEU on IWSLT Fr→En and En→De development sets with greedy decoding. For our value function, we use a 2-layer MLP with a ReLU nonlinearity.

The language model is a 1-layer recurrent language model with 512 LSTM hidden units. The image-caption retrieval model is a recently proposed VSE++ model (Faghri et al., 2018), with unidirectional 1-layer GRU with 512 hidden units and a single fully connected layer from 2048-dimensional ResNet features to 512-dimensional GRU hidden states. We report the performance of the pretrained models used in our experiments in Tables 1, 2 and 3.

|  | IWSLT | Multi30k |
|---|---|---|
| Fr→En | 34.05 | 26.80 |
| En→De | 21.94 | 18.56 |

Table 1: Translation performance of our pre-trained agents (BLEU)

| | |
|---|---|
| WikiText103 | 3.51 |
| MS COCO | 2.66 |
| Flickr30k | 2.85 |

Table 2: Development NLL of pretrained language models

|  | R@1 | R@5 | R@10 |
|---|---|---|---|
| Caption | 50.1 | 76.3 | 84.6 |
| Image | 35.7 | 65.3 | 75.9 |

Table 3: Retrieval results for our VSE++ model on Flickr30k test set.

**Training Details** When fine-tuning our agents, we perform learning rate annealing and early stopping based on Fr→De BLEU (communication performance) on the Multi30k development

set. We use Adam (Kingma & Ba, 2014) with initial learning rate of 0.001 and dropout (Srivas-tava et al., 2014) rate of 0.1. We grid search over learning rate schedule and reward coefficients $(\alpha_{\text{pg}}, \alpha_{\text{entr}}, \alpha_{\text{b}}, \beta_{LM}, \beta_G)$.

For our joint systems with policy gradient fine-tuning, we run every model three times with different random seeds and report averaged results (see Table 4).

**Baseline and Upper Bound**   Our main quantitative experiment has three baselines:

- Pretrained checkpoints (on IWSLT).

- Ensembling : Given Fr, we let Agent A generate $K$ English hypotheses with beam search, $\{\overline{\text{En}_j}\}_{j=1}^K$. Then, we let Agent B generate the German translation $\overline{\text{De}}$ using an ensemble of $K$ source sentences (Firat et al., 2016; Zoph & Knight, 2016).

- Fr→En fixed : We fix Agent A and only fine-tune Agent B using $\mathbb{L}_B$.

Meanwhile, we also train an NMT model of the same architecture and size directly on the Fr→De task in Multi30k Task 1. This serves as an upper bound on the Fr→De performance achievable with available data.

## 5   QUANTITATIVE RESULTS

|  | LM | Ranker | Fr→En | Fr→En→De |
|---|---|---|---|---|
| Pretrained | | | 27.18 | 16.30 |
| Ensembling | | | | 16.95 |
| Fr→En fixed | | | 27.18 | 22.37 |
| PG | No LM | | 12.38 (0.67) | 24.51 (1.48) |
| PG+LM | WikiText103 | | 21.63 (1.25) | 26.88 (0.12) |
| | MS COCO | | 25.05 (1.40) | 27.66 (0.34) |
| | Flickr30k | | 24.85 (1.14) | 27.60 (0.27) |
| | All | | 23.60 (1.05) | 27.67 (0.39) |
| PG+LM+G | No LM | ✔ | 14.20 (1.58) | 26.23 (1.08) |
| | WikiText103 | ✔ | 23.65 (1.91) | 27.87 (0.15) |
| | MS COCO | ✔ | 26.24 (0.28) | 27.86 (0.24) |
| | Flickr30k | ✔ | 25.99 (1.62) | 27.82 (0.41) |
| | All | ✔ | 24.75 (0.40) | 28.08 (0.73) |
| Fr→De | | | | 30.73 |

Table 4: Results in BLEU score on Multi30k Task 1. For our models using policy gradient fine-tuning, we report results averaged over three runs and provide standard deviations in brackets. PG: trained with vanilla policy gradient fine-tuning. PG+LM: trained with the "Englishness" constraint in reward. For MS COCO and Flickr30k, the LM was trained directly on image captions. PG+LM+G: trained with grounding loss as well as the LM loss. Fr→En: degree of intermediate language drift from English; lower indicates more drift. Fr→En→De: metric for communication accuracy; higher is better. All: LM was trained on all three datasets combined. Improvements of PG+LM+G over PG+LM were found to be significant in all cases, using the approximate randomization test for significance testing (Riezler & Maxwell III, 2005).

In Table 4, the top three rows are our baselines. The pretrained model performs relatively poorly on Fr→De, conceivably because it was pretrained on a different corpus, and Agent B was was given Agent A's output as source. Ensembling multiple English hypotheses for Agent B (row 2) gives negligible increase in Fr→De performance. When only Agent B is fine-tuned, we observe 6 BLEU score increase in Fr→De.

When the joint system is fine-tuned on German log likelihood with policy gradients (PG), we observe a large, 8 BLEU increase increase in Fr→De at the cost of a substanstial, 15 BLEU score drop in Fr→En. This clearly shows that optimizing on some external reward causes a drastic language drift.

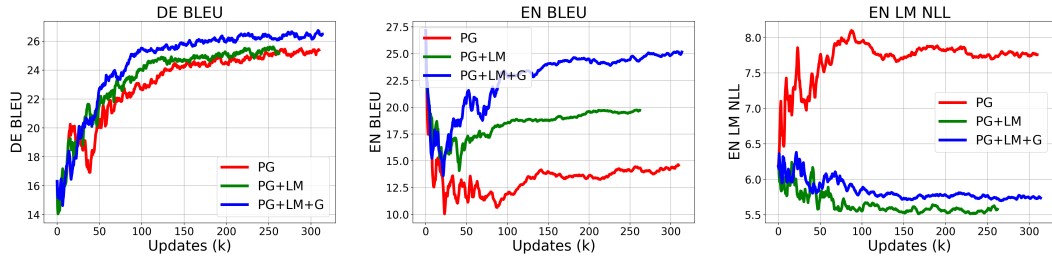

Figure 2: Learning curves for PG, PG+LM and PG+LM+G. En LM NLL curves show the NLL of English messages, computed by a language model trained on WikiText103. Lower En BLEU indicates more language drift, and higher En LM NLL indicates more language drift.

When the agent is trained with the "Englishness" constraint (PG+LM), we notice a significant improvement in Fr→En BLEU. When the LM is trained on WikiText103, a widely used language modelling dataset, we observe improvement of 9 BLEU scores. When the training corpus is closer to the target domain, such as MS COCO or Flickr30k, we see more than 10 BLEU score increase. Fr→De translation also improves by 2–3 BLEU scores.

However, we see the biggest improvements in performance when agents are trained using visual grounding feedback. This is particularly pronounced with the LM trained on WikiText103: introducing visual grounding leads to more than 2 BLEU score improvement in Fr→En, and 1 BLEU score improvement in Fr→De. We hypothesize that the "Englishness" constraint forces agents to communicate with correct syntax and fluency, while the image-caption retrieval model restricts the search space of languages to ones that are grounded by visual semantics. To see if grounding is really necessary, we train a stronger LM on all three datasets combined, but find this still leads to more language drift than using visual grounding: the PG+LM+G model with the LM trained on MS COCO outperforms this by 3 BLEU scores on Fr→En.

In Figure 2, we observe that vanilla PG fine-tuning quickly leads to highly "un-English" communication, as can be seen from a distinct increase in LM NLL. It is also worth noting that while PG+LM achieves better LM NLL than PG+LM+G, it gives much lower Fr→En BLEU score than the grounded model (PG+LM+G). This is another indication that simply encouraging naturalness is not enough; grounding is key.

A close investigation into the token statistics of each communication strategy reveals that PG fine-tuning causes the word frequency distribution to be flatter. The PG model has negative frequency difference values for the most frequent tokens, indicating that PG downweighs frequent words severely. On the other hand, PG+LM gives highly positive frequency differences, meaning that language modelling alone disproportionately emphasizes frequent tokens. Visual grounding keeps the token frequency distribution close to the original pretrained regimes. Analyzing the top-k most frequent

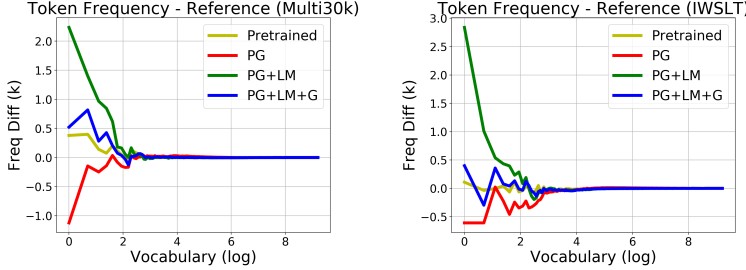

Figure 3: Token frequency analysis on three different models (PG, PG+LM, PG+LM+G) as well as the pretrained model before any fine-tuning (Pretrained). We show the word frequency curves (sorted in decreasing order) for each model, after subtracting the reference English frequency statistics (also sorted). Positive y values indicate higher frequency values than the English reference, and negative y values indicate lower frequency values than English. Note that y-axis is the frequency difference in thousands, and x-axis shows the vocabulary index (sorted with frequency) in log scale.

words shows that PG+LM disproportionately favors quotation marks, which are very common tokens in many language modelling datasets but occur rarely in Multi30k (see also Appendix A).

| | IWSLT | | | Multi30k | | |
|---|---|---|---|---|---|---|
| | unique | /sent | /all | unique | /sent | /all |
| Reference | 5,303 | 19.7 | 0.86 | 3,046 | 11.9 | 0.91 |
| Pretrained | 4,657 | 17.9 | 0.85 | 2,867 | 12.0 | 0.87 |
| PG | 4,933 | 13.6 | 0.56 | 3,197 | 9.2 | 0.65 |
| PG+LM | 3,819 | 14.6 | 0.61 | 2,438 | 10.9 | 0.78 |
| PG+LM+G | 4,327 | 15.7 | 0.74 | 2,550 | 10.7 | 0.84 |

Table 5: Additional token frequency analysis. unique: the number of unique English tokens used in the whole development set. /sent: the number of unique English tokens used per sentence. /all: (the number of unique English tokens / the number of all English tokens.)

Table 5 reinforces the finding that vanilla PG fine-tuning leads to flatter token frequency distributions, as the number of unique tokens used by PG is greater than that of the pretrained model. Meanwhile, PG+LM uses fewer tokens overall, signifying that it uses a relatively small set of tokens frequently.

Also note that PG, despite using a more diverse set of tokens, uses the smallest number of unique symbols per sentence (/sent) and overall (/all). This implies that PG communication is often repetitive. Introducing extra tasks seems to mitigate this, and the grounded model (PG+LM+G) learns a frequency distribution that most closely resembles the original distribution.

To gain further insight into the agents' communication protocols, we compare the degree of drift by part-of-speech. Table 6 shows that PG tends to ignore function words, such as periods and infinitives. Models trained with LM and grounding losses retain function words with much higher accuracy. PG fares relatively better with content words (nouns and verbs), but adding LM and grounding losses still outperform PG. Grounding leads to overall improvements in recall, particularly with content words.

Conceivably, when optimizing Agent A's policy on the communication task alone, it is more crucial to relay content information to Agent B, and this might cause agents to ignore syntactic conformity in the original intermediate language. We argue that LM and grounding reduces the space of intermediate languages to a much reasonable language space, facilitating learning.

## 6   QUALITATIVE RESULTS

In the first example of Table 7, it is clear that PG's English message has significantly diverged from English: it is highly repetitive ("table table table table table") and is missing some key content words such as "man" and "jacket". However, Agent B still generates the German word for 'man'. The grounded model's message (PG+LM+G) is distinctly the most fluent and semantically correct.

In the second example, observe that the PG Agent B misinterprets "talking talking a coach a coach" into "spricht mit einem spieler" (talking to a player). The PG+LM+G model again generates a flawless English sentence. Also note that it communicates both colors (red and white) successfully from French to German, while the other two models fail to do so.

| | Function words | | | Content words | | | |
|---|---|---|---|---|---|---|---|
| | TO | . | DT | Noun | Verb | Adj | Adv |
| PG | 0.22 | 0.36 | 0.57 | 0.38 | 0.17 | 0.32 | 0.26 |
| PG+LM | 0.55 | 0.84 | 0.72 | 0.39 | 0.18 | 0.21 | 0.25 |
| PG+LM+G | 0.62 | 0.88 | 0.74 | 0.43 | 0.26 | 0.33 | 0.29 |

Table 6: Exact-match word recall by POS-tag on IWSLT development set: when the English reference contains a word of a certain POS tag, how often does the agent correctly produces that word. TO: infinitive to, (.): period, DT: determiner, Noun: (NN, NNS, NNP, NNPS), Verb: (VB, VBD, VBG, VBN, VBP, VBZ), Adj: adjective (JJ, JJR, JJS), Adv: adverb (RB, RBR, RBS)

| | | |
|---|---|---|
| Ref | Fr | un vieil homme vêtu d'une veste noire regarde sur la table |
| | De | ein alter mann in einer schwarzen jacke blickt auf den tisch |
| | En | an old man wearing a black jacket is looking on the table |
| En | PG | a old teaching black watching on the table table table table table table |
| | +LM | a old man in a jacket looking on the table . " " |
| | +G | an old man in a black jacket looking on the table . |
| De | PG | ein älterer mann in einem schwarzen hemd schaut auf den tisch . |
| | +LM | ein alter mann in einer jacke beobachtet einen tisch . |
| | +G | ein älterer mann in einer schwarzen jacke schaut auf den tisch . |
| Ref | Fr | un joueur de football américain en blanc et rouge parle à un entraîneur . |
| | De | ein rot-weiß gekleideter footballspieler spricht mit einem trainer . |
| | En | a football player in red and white is talking to a coach . |
| En | PG | a player football american football american and red talking talking a coach |
| | +LM | a player of white and red talking to a coach . " " " |
| | +G | a football player in white and red talking to a coach . |
| De | PG | ein footballspieler spricht mit einem spieler in einem roten trikot . |
| | +LM | ein weiß gekleideter fußballspieler spricht zu einem trainer . |
| | +G | ein fußballspieler in einem rot-weißen trikot spricht mit einem trainer . |

Table 7: Two random examples from Multi30k development set with different models (PG, PG+LM, PG+LM+G). The top three rows list the ground truth sentences, the middle three rows are the English messages sent by the Fr→En agent, and the bottom three rows show the German output from the En→De agent. We also show the corresponding images, which were only used to train the image-caption retrieval modal.

| | |
|---|---|
| Fr src | un enfant assis sur un rocher. |
| En ref | a child sitting on a rock formation. |
| En hyp | a punk sitting sitting on on a broken |
| De ref | ein kind sitzt auf einem felsen . |
| De hyp | ein kind sitzt auf einem felsen . |
| Fr src | un petit enfant est assis à une table, en train de manger un goûter. |
| En ref | a toddler is sitting at a table eating a snack . |
| En hyp | a punk sits sitting sitting next next a airline |
| De ref | ein kleines kind sitzt an einem tisch und isst einen snack . |
| De hyp | ein kind sitzt an einem tisch und liest ein buch . |

Table 8: Evidence of token flipping in the PG model.

We observe some instances of token flipping with the PG model. For example, one particular PG model uses "punk" to describe "child" (see Table 8). As no occurrence of "punk" in any training data is associated with "child", the agents must have acquired this new meaning assignment during fine-tuning. Among 35 examples in Multi30k development set where the English reference contains "child", the model uses "punk" 15 times, indicating this is no random phenomenon. We show similar examples from the PG+LM model in Appendix B. We did not observe such examples with the PG+LM+G model.

# 7 CONCLUSION

In this paper, we show that language drift happens when fine-tuning natural language agents with some external (non-linguistic) reward using policy gradients, and propose a few approaches to avoid this. Most importantly, we find that simply encouraging "naturalness", e.g. via adding a language model log likelihood to the reward, does not lead to the desired consequences. Instead, we contend that grounding is what we need to avoid language drift. Our empirical results show that grounding leads to best communication performance (highest Fr→De BLEU), while also showing least signs of language drift (highest Fr→En BLEU). Analyzing token frequencies in exchanged messages reveals that pure PG finetuning tends to learn flatter token distributions, and encouraging naturalness disproportionately emphasizes frequent tokens, while the grounded model best retains the original token frequencies.

ACKNOWLEDGMENTS

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

## A    FREQUENCY ANALYSIS

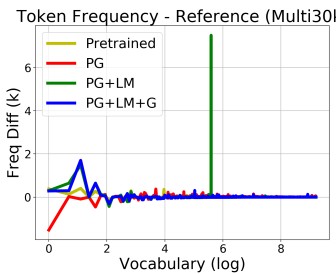
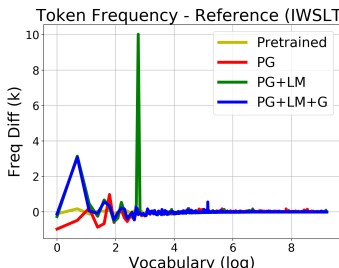

Figure 4: Token frequency analysis similar to Figure 3, but with the x-axis fixed to the token indices sorted with respect to English reference, in decreasing order.

In Figure 4, where the x-axis is fixed to the token indices sorted with respect to the English reference, we observe that the PG+LM model does indeed favor one word particularly strongly. From investigating top-k most frequent tokens in each model, we find that quotation mark is the most common token for PG+LM in both datasets we experimented with. It is plausible that quotation marks occur with high frequency in language modelling datasets, causing them to be disproportionately overweighed during fine-tuning.

| IWSLT | |
| --- | --- |
| Reference | , . the and to of a that i in is it you we 's this " |
| Pretrained | , the . to of and a i that in it we you 's is this " was |
| PG | a the and , . in i " this of to is we you ? that not for |
| PG+LM | " the , of . and in a to this is i es you for we that with |
| PG+LM+G | the , . of a and to in is i this es we for that you at what |
| Multi30k | |
| Reference | a . in the on of with and is man two woman to are people at an |
| Pretrained | a . the in of on with and , man @-@ is to woman two people at white |
| PG | a in the on and with . man water red two woman street blue city its people white |
| PG+LM | " a the . in of on with and man , to two woman his people es young |
| PG+LM+G | a the . in of on with and man at es to , woman two his people is |

Table 9: Top 20 most frequent tokens in English reference (Reference) or the output from Fr→En models.

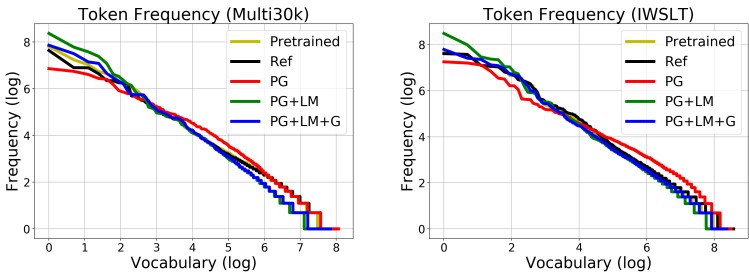

Figure 5: Token frequency curves (before subtracting the reference frequencies). Both x (vocabulary index) and y (frequency) axes are in log scale.

In Figure 5, we show the token frequency curves before subtracting the reference frequencies. Similarly to Figure 4, we observe that the PG model discourages frequent (mostly functional) words, while the PG+LM model excessively prefers frequent words.

## B EVIDENCE OF TOKEN FLIPPING IN THE PG+LM MODEL

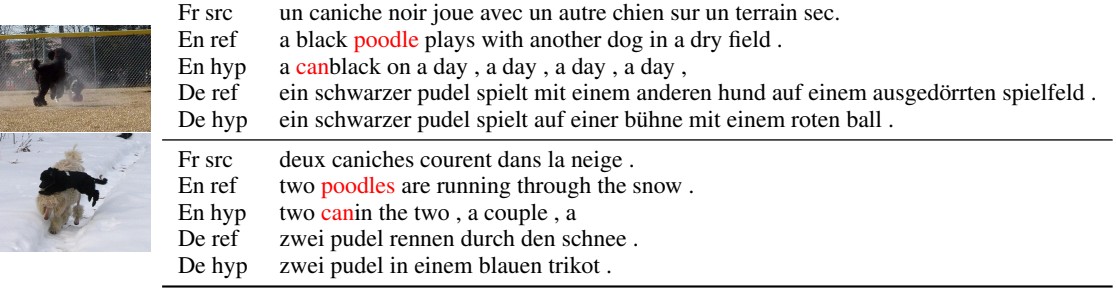

| | |
|---|---|
| Fr src | un caniche noir joue avec un autre chien sur un terrain sec. |
| En ref | a black poodle plays with another dog in a dry field . |
| En hyp | a canblack on a day , a day , a day , a day , |
| De ref | ein schwarzer pudel spielt mit einem anderen hund auf einem ausgedörrten spielfeld . |
| De hyp | ein schwarzer pudel spielt auf einer bühne mit einem roten ball . |
| Fr src | deux caniches courent dans la neige . |
| En ref | two poodles are running through the snow . |
| En hyp | two canin the two , a couple , a |
| De ref | zwei pudel rennen durch den schnee . |
| De hyp | zwei pudel in einem blauen trikot . |

Table 10: Evidence of token flipping in the PG+LM model.

Similar to Table 8, we find evidence of token flipping for the PG+LM model, where the agents use "can@@" (@@ is a subword BPE token marker) to mean "poodle". This shows that language drift still happens even when a language model is used.

