# OpenReview forum: "Countering Language Drift via Grounding"
_ICLR.cc/2019/Conference_

### Official Review · AnonReviewer1 · 2018-10-31
**Important topic, but uncertain about framing and significance of results**

**Rating:** 6
**Confidence:** 4

**Review:**

This paper poses and addresses the problem of language drift in multi-agent communication paradigms. When two pretrained natural-language agents are jointly optimized to communicate and solve some external non-linguistic objective, their internal communication often diverges to a code-like, unnatural communication system. This paper solves this “language drift” problem by requiring that the messages between agents be usable as inputs to an image caption retrieval system. They demonstrate that the jointly optimized agents perform best when regularized in this manner to prevent language drift.

1. Framing: I’m uncertain about the framing of this paper. The authors pose the problem of “language drift,” which is indeed a frequent problem in multi-agent communication tasks where the principle supervision involves non-linguistic inputs and outputs. They then design a three-language MT task as a test case, where the inputs and outputs are both linguistic. Why attack this particular task and grounding solution? I can imagine some potential goals of the paper, but also see more direct ways to address each of the potential goals than what the authors have chosen:
1a. Study how to counter language drift in general — why not choose a more intuitive two-agent communication task, e.g. navigation, game playing, etc.?
1b. Study how to counter language drift in the MT task — aren’t there simpler solutions to prevent language drift in this particular task? e.g. require “cycle-consistency” – that it be possible to reconstruct the French input using the French output? Why pick multimodal grounding, given that it imposes substantial additional data requirements?
1c. Build a better/more data-efficient machine translation system — this could be an interesting goal and suitable for the paper, but this doesn’t seem to be the framing that the authors intend.

2. Interpretation of first results:
2a. Thanks for including standard deviation estimates! I think it’s also important that you do some sort of significance testing on the comparison between PG+LM+G and PG+LM performance for Fr->En->De — these numbers look pretty close to me. You could run e.g. a simple sign test on examples within each corpus between the two conditions.
2b. It would also be good to know how robust your results are to hyperparameter settings (especially the entropy regularization hyperparameter).

3. Token frequency results: These are intriguing but quite confusing to me!
3a. How sensitive are these results to your entropy regularization setup? How does PG behave without entropy regularization?
3b. Table 6 shows that the PG model has very different drift for different POS categories. Does this explain away the change in the token frequency distribution? What do the token frequency effects look like for PG within the open-class / content word categories (i.e., controlling for the huge difference in closed-class behavior)?

4. Minor comments:
4a. There’s a related problem in unsupervised representation learning for language. Work on VAEs for language, for example, has shown that the encoder often collapses meaning differences in the latent representation, and leans on an overly powerful decoder in order to pick up all of the lost information. It would be good to reference this work in your framing (see e.g. Bowman et al. (2015)).
4b. In sec. 3.1 you overload notation for R. Can you subscript these so that it’s especially clear in your results which systems are following which reward function?
4c. Great to show some qualitative examples in Table 7 — can you explicitly state where these are from (dev set vs. test set?) and whether they are randomly sampled?

References:
Bowman et al. (2015). Generating sentences from a continuous space. https://arxiv.org/abs/1511.06349

---

> ### Comment · Area_Chair1 · 2018-11-08
> **An additional extremely simple baseline?**
>
> Hello, area chair here.
>
> I agree with the reviewer here on point 1: this setting is a bit artificial, and it seems that there are much simpler ways to prevent language drift. In addition to the "cycle consistency" loss, it seems even easier to occasionally sample batches of French-English data, and train on these batches using the original MLE objective that was used to pre-train the model. Do the methods in this paper have any advantages over this simple auxiliary objective?
>
> I could see an argument that maybe grounding in images is more cognitively plausible if you were trying to simulate how babies learn or something like that, but in this case the experimental setting of having two MT systems is far from cognitively plausible in the first place.

---

> > ### Author Response · Authors · 2018-11-09
> > **Author response**
> >
> > Dear AnonReviewer1 and AC,
> >
> > Thank you both for your constructive feedback! We think there is a slight misunderstanding here: the reason we have chosen this setup is exactly because this particular task and setup directly addresses the problem of language drift, in a way where the semantics stays identical while the communication channel gets only extrinsic reward (i.e., the meaning is exactly the same for all languages and modalities). In addition, every single utterance has very clear and very well-known metrics, in the shape of BLEU and NLL/perplexity, allowing us to measure performance at every single step.
> >
> > We would very much welcome any suggestions for other tasks where this setup would be possible, and where data is available, but we think that AnonReviewer1's suggestions (while of course very welcome) do not satisfy these criteria: other two-agent communication tasks such as navigation or game-playing have neither clearly defined metrics nor easily available NL data. Hence, we do not think our setup is artificial, but ideal for understanding the problem as best as we possibly can.
> >
> > As for auxiliary objectives or other solutions for preventing drift:
> > - If cycle-consistency means a French-French auto-encoder with English intermediary, that is a much weaker setting than French-German with English intermediary, because there is no way for the agents to focus on superficial information and really only the meaning of the sentence is at stake. Our setting is much more difficult than this sort of cycle-consistency, as the focus is on the semantics, which is why we want to avoid language drift in the first place. In other words, cycle-consistency is a special case of what we've done, we do not expect any different result from it.
> > - Occasionally sampling batches of French-English data with the original MLE objective does not prevent drift, unless we do that so often that we undo the advantages of fine-tuning. Note that performance is much better than training simply with an MLE objective, due to fine-tuning. Occasionally training with MLE will avoid some drift but at the expense of performance improvements. Our work clearly shows that the alternative solution of adding language model constraints is insufficient.
> >
> > If you like, we would be happy to train French-French and German-German autoencoders, with English intermediates, and show that the same results hold as what we report for the harder French-German case. Similarly, we would be happy to add experiments showing that occasionally training on batches with MLE does not work as well as our proposed solution.
> >
> > With regard to cognitive plausibility of the three-way translation task, we disagree: In 2009, the United States Census Bureau reported that about 20 percent of Americans speak a language other than English at home. Therefore we would like to point out that this three-way translation task, using English as an intermediate language, is not only cognitively plausible but a reality for one fifth of Americans.
> >
> > A useful interpretation of our setup is that Agent A is communicating their (French) thoughts in English so that Agent B understands the intended message in their (German) thoughts. In a sense our setup is more plausible than the cycle-consistent autoencoder, because no two people speak exactly the same language or have exactly the same thoughts.

---

> > > ### Comment · Area_Chair1 · 2018-11-19
> > > **Concerns about applicability of method**
> > >
> > > Hello,
> > >
> > > Thank you for the comprehensive response.
> > >
> > > > We would very much welcome any suggestions for other tasks where this setup would be possible, and where data is available, but we think that AnonReviewer1's suggestions (while of course very welcome) do not satisfy these criteria: other two-agent communication tasks such as navigation or game-playing have neither clearly defined metrics nor easily available NL data.
> > >
> > > I'm concerned by this: does this mean that there are no tasks for which the proposed method here is actually applicable? If it is such as struggle to come up with an appropriate test bed, then it seems that the underlying motivation itself is lacking.
> > >
> > > > With regard to cognitive plausibility of the three-way translation task, we disagree: In 2009, the United States Census Bureau reported that about 20 percent of Americans speak a language other than English at home.
> > >
> > > Right, but the language they learn at home is not learned by translating between two other languages, like the task presented here. Rather, it is learned by grounding in inputs (acquired through vision) from the world around. To see an example of a more cognitively plausible model, please see, for example "Learning Words from Sights and Sounds: a Computational Model" (Roy and Pentland, Cognitive Science 2002)
> > >
> > > > A useful interpretation of our setup is that Agent A is communicating their (French) thoughts in English so that Agent B understands the intended message in their (German) thoughts. In a sense our setup is more plausible than the cycle-consistent autoencoder, because no two people speak exactly the same language or have exactly the same thoughts.
> > >
> > > I can understand that this might be a proxy for the task of "thoughts" in one person's brain to "thoughts" in another, but then the question becomes, how do you get the supervision for "thoughts -> English" in the first place? The point of this paper is preventing language "drift", which is divergence from an original model that has been trained on "thoughts -> English", but if the whole method it predicated on having this original model, then it's not clear to me why this model is interesting and/or useful.
> > >
> > > Sorry if this sounds somewhat negative, but I'd really like to understand the motivation. Thanks!

---

> > > > ### Author Response · Authors · 2018-11-21
> > > > **Author response**
> > > >
> > > > Dear AC,
> > > >
> > > > Thanks for the detailed response! We appreciate the feedback.
> > > >
> > > > > I'm concerned by this: does this mean that there are no tasks for which the proposed method here is actually applicable? If it is such as struggle to come up with an appropriate test bed, then it seems that the underlying motivation itself is lacking.
> > > >
> > > > Ah, we see what you mean: no, this does not mean our method is not applicable to any tasks---in fact, it is very generally applicable to any task that involves fine-tuning of language generation components/decoders. We think our current test bed of three-way translation is most appropriate for examining and understanding exactly what happens at each step. We think it is important to really understand what is going on, rather than just showing an improvement on an arbitrary generation task without knowing (and being able to measure) where that improvement comes from. Our methods can be applied to any fine-tuning of language generation, including in MT, but also summarization, style transfer, dialogue, et cetera. We respectfully disagree with the notion that we struggled to come up with an appropriate test bed: to reiterate, we think our current setup is the most appropriate for understanding this problem as best as possible.
> > > >
> > > > > With regard to cognitive plausibility of the three-way translation task, we disagree: In 2009, the United States Census Bureau reported that about 20 percent of Americans speak a language other than English at home.
> > > > Right, but the language they learn at home is not learned by translating between two other languages, like the task presented here. Rather, it is learned by grounding in inputs (acquired through vision) from the world around. To see an example of a more cognitively plausible model, please see, for example "Learning Words from Sights and Sounds: a Computational Model" (Roy and Pentland, Cognitive Science 2002)
> > > >
> > > > As we said in our previous response, one can think of the setup as using language as an intermediary between the agents “mental states” (for which we use language in order to be able to exactly measure things, but they could be represented differently). Thank you for the Roy & Pentland suggestion, we are aware of this work and we obviously agree that grounding is super important: however, a large part of learning to speak a language also revolves around making sure that the other agent correctly understood what you said, in addition to being grounded - which is much closer to our setup. In fact, there has been a lot of debate in the cogsci community around which of these two is more important for meaning (grounding or communication), see e.g. the work of Louwerse and to a lesser extent Barsalou, and our methods might shed some light on this question in the longer term.
> > > >
> > > > > A useful interpretation of our setup is that Agent A is communicating their (French) thoughts in English so that Agent B understands the intended message in their (German) thoughts. In a sense our setup is more plausible than the cycle-consistent autoencoder, because no two people speak exactly the same language or have exactly the same thoughts.
> > > > I can understand that this might be a proxy for the task of "thoughts" in one person's brain to "thoughts" in another, but then the question becomes, how do you get the supervision for "thoughts -> English" in the first place? The point of this paper is preventing language "drift", which is divergence from an original model that has been trained on "thoughts -> English", but if the whole method it predicated on having this original model, then it's not clear to me why this model is interesting and/or useful.
> > > >
> > > > While our current setup is (French) thoughts -> English -> (German) thoughts, our findings apply broadly to any task that involves fine-tuning a language generation module with an external reward signal. For example, one could take a dialogue system pretrained on ground truth utterances, and fine-tune it using some ultimate metric that they care about: e.g. engagingness, diversity, consistency. In doing so, our findings suggest that vanilla PG fine-tuning might result in language drift. Alternatively, imagine a natural language based navigation system. This might be pretrained on a small supervised dataset, but has to be fine-tuned on more abstract metrics, e.g. user happiness, avoiding traffic. We believe that there are many such tasks where fine-tuning a language generation module is desired, which makes our model both interesting and useful.

---

> > > > ### Comment · AnonReviewer2 · 2018-12-13
> > > > **Is the solution consistent with the motivation?**
> > > >
> > > > I do agree AC's point that humans learn language via grounding in inputs form the physical world. It indeed is impractical that two persons speaking different languages understand each other via the "third" language (both of them might can only speak in ONE language). I think the motivation/intuition behind this paper is good, but the solution is a bit artificial. If the goal is to explore how humans learn languages or understand others speaking different languages, I'd prefer to approach the problem via cognitive models. Simply solving the problem (e.g., improve BLEU score) via some computational techniques might violate the initial motivation.

---

> > > > > ### Author Response · Authors · 2018-12-13
> > > > > **Solution is consistent, and measurable whereas alternatives are not**
> > > > >
> > > > > Dear AnonReviewer2,
> > > > >
> > > > > Thank you for your comment! We really appreciate you taking the time!
> > > > >
> > > > > As we explained in response to this point below, the setup is the way it is for measurability: if you want to really understand this problem, the ideal setup is three languages, where the "input language" and "output language" can be seen as a way to represent "thoughts" communicated via the intermediate language, the semantic content is identical and the metric (BLEU) is well-understood. If the agents had the same language, it would be possible to cheat, which is not possible with our setup. We could have done the same thing with a single language as a three-way auto-encoder (e.g. using COCO), but this would only make the problem more difficult to examine and measure.
> > > > >
> > > > > To reiterate, the goal is to examine and address the problem of language drift, which is not what you appear to think our goal is. We show how grounding can be useful in avoiding language drift, by imposing important semantic (rather than syntactic) constraints. This is indeed a strong motivation for why grounding helps in how humans learn language, but that is not the point we are arguing here (although we fully agree with you of course).
> > > > >
> > > > > With regard to your review: we would appreciate it if you can reconsider your original score, which was based on a weakness we have since thoroughly addressed.
> > > > >
> > > > > Can you please explain in more detail how we should approach the problem with cognitive models, why this would be more measurable, and why that solution is more consistent with the motivation? Again, we really appreciate your time and very valuable feedback!

---

> > > > > > ### Comment · AnonReviewer2 · 2018-12-14
> > > > > > **the third language is necessary?**
> > > > > >
> > > > > > Looks like my comments confuse you a bit. I did not mean the same language. For example, person A can only speak Chinese, and person B can only speak Spanish. How they communicate with each other? No one understand the third language. This is very common in human's learning process.

---

> > > > > > > ### Author Response · Authors · 2018-12-14
> > > > > > > **the third language is the intermediate**
> > > > > > >
> > > > > > > We apologize if we misunderstood. To clarify: the third language here is English: we have agent A who speaks French, agent B who speaks German and in this case, they have to learn to communicate in English so that they understand each other in their original languages. How does one go about this? We show that if you do not ground the English intermediate, it drifts and you cannot expect the intermediate to make sense. We are saying that if it was EN->EN->EN or DE->EN->DE or whatever other combination, it would be harder to measure than FR->EN->DE, which does  exactly what we wanted to check (i.e., semantically identical content without syntactic overlap). Does that make sense?

---

> > > > > > > > ### Comment · AnonReviewer2 · 2018-12-14
> > > > > > > > **The key thing is metric!!**
> > > > > > > >
> > > > > > > > I understand that you want something there to measure if they understand each other. The key thing is if such metric is cognitive plausible or unique or the best? I think Review 1 also concern about this. And, this is why I am asking if the third language is necessary. For example, our task is navigation. We let two persons speaking different languages corporate with one another to complete a task. If the goal is "go to kitchen", they talk with each other to figure out strategies. If they do complete the task, that means they understand each other, this is a kind of metric. We just concern if this setting makes a lot sense or is there any better ways. If our goal is to improve the BLEU score, whatever setting is fine, but if the goal is to explore how to advance AGI, we might need to care about framing.

---

> > > > > > > > > ### Author Response · Authors · 2018-12-14
> > > > > > > > > **Exactly!**
> > > > > > > > >
> > > > > > > > > We agree with you that the metric is the key, and that is precisely why we proposed this setup as a good way (not necessarily "the" way) to tackle this problem. Please also note that the goal is not to improve the BLEU score: we do not fine-tune for BLEU score but for the negative log-likelihood of the output sequence, see section 3.2.
> > > > > > > > >
> > > > > > > > > We do not claim to have come up with "the only way" to tackle this issue, but only that this setup of bridged translation is a good setup in which we can understand the emergent language as well as how such a language changes over time. Machine translation provides us with a way to check (1) whether the intermediate protocol is effective by checking the BLEU score between French to German, and also (2) whether the intermediate language stays English and describes the same thing in the original French or German sentence again by BLEU score. We argue that it is not easy to come up with a setup in which both of these are satisfied. For instance in your example of navigation, (1) is straightforward to check by simply seeing whether the agent arrived at the kitchen, but (2) is less so as there are many different ways to describe "go to the kitchen" and it is not trivial to check whether a particular emergent protocol's sentence refers to one of many (if not infinitely many) variants of "go to the kitchen". In our case, which is quite special, we can precisely measure both - one metric is completing the task, and the other is completing the task the right way (without drift).
> > > > > > > > >
> > > > > > > > > Let us re-emphasize that we are not pushing this setup as the only or best cognitively plausible setup in which emergent language and its drift is studied. It is "a" good setup in which we can study this phenomenon, and we have conducted thorough experimentation and analyses under this setup (not to mention conducting additional analyses at the reviewers' request).

---

> ### Author Response · Authors · 2018-11-19
> **Author response**
>
> Dear AnonReviewer1,
>
> Thank you for your very detailed feedback, we appreciate your efforts! Please see our response below.
>
> Response to 1(a, b, c) : please see the separate response to AnonReviewer1 and AC below.
>
> (2a) We agree that Fr->En->De results are pretty close between PG+LM and PG+LM+G. Our qualitative analyses, on the other hand, strongly indicate that PG and PG+LM learn a biased token distribution, and that visual grounding is key to retaining the original token distribution.
>
> (2b/3a) We performed a thorough grid search over entropy regularisation, and selected \alpha_entr = 0.001 based on Fr->En->De performance on the validation set. A lower value (or no regularisation) would sometimes lead to models being stuck with only a few vocabulary words being used (therefore higher variance in the overall results). When a larger value is used, the Fr->En agent uses excessively many tokens, and the models do not converge.
>
> (3b) Yes, this seems to affect the token frequency distribution. Vanilla PG training significantly discourages function words: period and comma are ranked much lower for PG than for English reference (see Table 9). Meanwhile, PG encourages content/open-class words. In Table 9, content words such as “red”, “blue”, “city”, “white” are ranked high for PG, much more so than other models.
>
> (4a) This is a very interesting point. We mentioned this in the related work section in our revision.
>
> (4b) Thanks for the suggestion. This is fixed in the revision.
>
> (4c) These are random samples from the development set. We modified the captions to reflect this.

---

> > ### Comment · AnonReviewer1 · 2018-11-21
> > **Reviewer response**
> >
> > Hi authors—thanks for your prompt and detailed response!
> >
> > I'll hold my response to #1 for the other thread below. Regarding other points:
> >
> > (2a) I see your qualitative results, but I really think you need to back up these main claims with quantitative tests. You claim in your defense of the framing / model setup that this is a good design because BLEU/NLL are well-defined and extremely informative about the degree to which a model is succeeding on the "external" fine-tuning task. If that's true, a significance test between PG+LM and PG+LM+G should also be very informative! (Consider e.g. a Wilcoxon sign test on paired outputs for each target sentence, compared between models.)
> >
> > (3b) Your story about content words in PG vs. PG+LM+G makes intuitive sense — to the extend that Agent B can re-construct the necessary syntactic relations between content words, the closed-class words originally cuing those syntactic relations can be dropped. I'm not sure if I'm supposed to learn something else from the frequency distribution differences, or if that's it? (This is more about my uncertainty than a concrete criticism of your paper. Unless I'm missing something, I might just suggest leading with the closed-class vs. open-class comparisons, since these are totally easy to understand — much less mysterious than the more general statements about frequency changes. But epistemic status is not very high here.)

---

> > > ### Author Response · Authors · 2018-11-26
> > > **Author response**
> > >
> > > Dear AnonReviewer1,
> > >
> > > Thanks for your constructive feedback!
> > >
> > > > a significance test between PG+LM and PG+LM+G should also be very informative!
> > >
> > > Thanks for your suggestion. Per your request, we conducted pair-wise statistical testing between LM and LM+G for each setup (see Riezler and Maxwell III, 2005 for details), by taking the model instance with the median Fr->En->De performance for each model. In all five cases considered (No LM, WikiText103, MS COCO, Flickr30k, All), we found the difference between LM and LM+G to be statistically significant. All comparisons were p<0.01, except the LM=All case (p<0.02) which had substantially more data. We added this in the latest version of the draft.
> > >
> > > > I'm not sure if I'm supposed to learn something else from the frequency distribution differences, or if that's it?
> > >
> > > The frequency difference curves are interesting as they show that PG and PG+LM finetuning result in token frequency distributions that are quite different from English. Most frequent words are closed-class words, but not all of them (see words “man”, “woman” and “people” in the second example of Table 9). On a higher level, this plot shows that PG+LM+G results in a token frequency distribution closest to that of the reference English.
> > >
> > > On the other hand, Table 6 shows a finer grained analysis of word recall by POS-tag, where pure PG fine-tuning tends to ignore closed-class words, and adding grounding loss helps mitigate this across a range of classes.

---

### Official Review · AnonReviewer3 · 2018-11-01
**Is BLEU the right metric?**

**Rating:** 6
**Confidence:** 4

**Review:**

The paper presents an approach to refining a translation system with grounding (in addition to LM scores) in the loop to manage linguistic drift.  The intuition is straightforward and results are clearly presented, but the gains are unfortunately much weaker than I would have hoped for.

The results for both Fr-En and Fr-En-De only show very small gains for adding grounding, often with PG+LM results being within 1 std-dev of the PG+LM+G results.  Otherwise, the results are quite nice with interesting increases in linguistic diversity.  This leads me to wonder if this approach would show more gains with a human evaluation rather than BLEU score.

What is the performance of PG+G without the +LM?

Minor -- In Fig 2, should the green line (PG+LM) have continued climbing to >21 BLEU?

---

> ### Author Response · Authors · 2018-11-19
> **Author response**
>
> Dear AnonReviewer3,
>
> Thank you for your helpful comments! Please see our response below.
>
> - Is BLEU the right metric? : Given the availability of ground truth English references, we believe BLEU is the best metric we have, as it is clearly interpretable and well-understood. BLEU of course also has weaknesses, and we agree that a human evaluation would be very interesting. In this setting, however, we believe BLEU is sufficient for making our argument.
>
> - Gains for adding grounding are small (within 1 std dev): We agree that the results for PG+LM and PG+LM+G are often close in BLEU score. Our qualitative analyses, on the other hand, strongly indicate that PG and PG+LM models learn a biased token distribution: namely, PG finetuning ignores the content words, while the PG+LM finetuning excessively encourages them. We find that visual grounding is key to retaining the original token distribution.
>
> - The results of PG+G without +LM is shown in Table 4, with PG+LM+G with “LM=None”. In hindsight we see that we should have made this clearer: we updated the table to make this stand out more. The PG+G model, although outperformed by PG+LM and PG+LM+G, still produces less drift than the vanilla PG finetuning alone.
>
> - Fig. 2, would PG+LM continue to over 21? : We used the same early stopping criteria based on communication performance (Fr->En->De BLEU) for all our models. Hence this model was early stopped at that point.

---

### Official Review · AnonReviewer2 · 2018-11-05
**Countering Language Drift via Grounding**

**Rating:** 6
**Confidence:** 3

**Review:**

Summary:
This paper tries to verify a hypothesis that language grounding DO help to overcome language drift when two agents creating their own protocol in order to communicate with each other. There are several constraints to enforce: 1) naturalness, say "Englishness", 2) grounded in visual semantics. The experiments prove that both constraints help the most (say, BLUE score). 1) w/o 2) restricts the vocabulary into a small set with the most frequent words, while 1) with 2) can resemble the original distribution.

Strength:
- How to make the protocol automatically created by two agents much explainable/meaningful is a very interesting topic. This paper explores plausible constraints to reach this goal.

Weakness:
- Visual grounding task brings more data there. To fairly compare, I hope to add one more baseline PG+LM+G_text, where G_text simply means to use text data (captions) alone, i.e., without visual signals.

---

> ### Author Response · Authors · 2018-11-19
> **Author response**
>
> Dear AnonReviewer2,
>
> Thank you for your constructive feedback! We agree that making multi agent communication more interpretable is both interesting and important.
>
> - Grounding “brings more data” :  thanks for pointing this out. We added this stronger baseline where the LM was trained on both WikiText103 and the captions from Flickr30k and MS COCO datasets (see Table 4, LM=All), and found that introducing visual grounding still outperforms this baseline.

---

> > ### Author Response · Authors · 2018-12-13
> > **Update score**
> >
> > Dear AnonReviewer2,
> >
> > Considering that we have addressed the weakness you raised by adding a much stronger baseline which we still beat by a considerable margin, would you please consider revising your scores accordingly? Thank you again for your valuable reviews and for considering our responses and revisions.

---

### Public Comment · (anonymous) · 2018-12-17
**Some Questions on Implementation**

Hi Authors,

This is a very interesting setting and line of work, and I am working on reproducing this. Before I do, there are a couple of questions in my mind:
1. For simple PG baseline, the reward for Agent A is log p(De_i | \bar{En_i}), which if I understand correctly, the log-likelihood of all sentence. But this reward would add bias to the shorter sentence, since the log-likelihood of the sentence is the sum of the loglikelihood of each words given context. So would it be a better choice here if the reward is a length normalized log-likelihood?
2. During RL training, which method did you use to get the action from agent A? Is it greedy decoding or sampling with some temperature?

---

> ### Author Response · Authors · 2018-12-17
> **Response to questions**
>
> Hi,
>
> Thanks for your interest in the paper! Please see the response below.
>
> 1. Yes, you're right. We use a length normalized log-likelihood (of the sentence) as the reward to avoid this behaviour.
>
> 2. At each timestep of Agent A's decoder, we sample an action with temperature=1.

---

### Meta-Review · Area_Chair1 · 2018-12-13
**Reasonably complete experiments, but motivation of method is still not clear.**

**Confidence:** 4
**Recommendation:** Reject

**Metareview:**

This paper proposes a method to resolve "language drift," where a pre-trained X->language model trained in an X->language->Y pipeline drifts away from being natural language. In particular, it proposes to add an auxiliary training objective that performs grounding with multimodal input to fix this problem. Results are good on a task where translation is done between two languages.

The main concern that was raised with this paper by most of the reviewers is the validity of the proposed task itself. Even after extensive discussion with the authors, it is not clear that there is a very convincing scenario where we both have a pre-trained X->language, care about the intermediate results, and have some sort of grounded input to fix this drift. While I do understand the MT task is supposed to be a testbed for the true objective, it feel it is necessary to additionally have one convincing use case where this is a real problem and not just the artificially contrived. This use case could either be of practical use (e.g. potentially useful in an application), or of interest from the point of view of cognitive plausibility (e.g. similar to how children actually learn, and inspired by cognitive science literature).

A concern that offshoots from this is that because the underlying idea is compelling (some sort of grounding to inform language learning), a paper at a high-profile conference such as ICLR may help re-popularize this line of research, which has been a niche for a while. Normally I would say this is definitely a good thing; I think considering grounding in language learning is definitely an important research direction, and have been a fan of this line of work since reading Roy's seminal work on it from 15 years ago. However, if the task used in this paper, which is of questionable value and reality, becomes the benchmark for this line of work I think this might lead other follow-up work in the wrong direction.  I feel that this is a critical issue, and the paper will be much stronger after a more realistic task setting is added.

Thus, I am not recommending acceptance at this time, but would definitely like the authors to think hard and carefully about a good and realistic benchmark for the task, and follow up with a revised version of the paper in the future.